# Motifs in Natural Products as Useful Scaffolds to Obtain Novel Benzo[*d*]imidazole-Based Cannabinoid Type 2 (CB2) Receptor Agonists

**DOI:** 10.3390/ijms241310918

**Published:** 2023-06-30

**Authors:** Analia Young Hwa Cho, Hery Chung, Javier Romero-Parra, Poulami Kumar, Marco Allarà, Alessia Ligresti, Carlos Gallardo-Garrido, Hernán Pessoa-Mahana, Mario Faúndez, Carlos David Pessoa-Mahana

**Affiliations:** 1Pharmacy Department, Faculty of Chemistry, Pontificia Universidad Católica de Chile, Vicuña Mackenna 4860, Santiago 7820436, Chile; aycho@uc.cl (A.Y.H.C.); hchung@uc.cl (H.C.); mfaundeza@uc.cl (M.F.); 2Organic Chemistry and Physical Chemistry Department, Faculty of Chemical and Pharmaceutical Sciences, Universidad de Chile, Olivos 1007, Santiago 7820436, Chile; jhromero@uc.cl (J.R.-P.); hpessoa@ciq.uchile.cl (H.P.-M.); 3National Research Council of Italy, Institute of Biomolecular Chemistry, 80078 Pozzuoli, Italy; p.kumar@icb.cnr.it (P.K.); mallara@icb.cnr.it (M.A.); aligresti@icb.cnr.it (A.L.)

**Keywords:** cannabinoid receptor, natural products, benzimidazoles, CB2 agonists, synthesis

## Abstract

The endocannabinoid system (ECS) constitutes a broad-spectrum modulator of homeostasis in mammals, providing therapeutic opportunities for several pathologies. Its two main receptors, cannabinoid type 1 (CB1) and type 2 (CB2) receptors, mediate anti-inflammatory responses; however, their differing patterns of expression make the development of CB2-selective ligands therapeutically more attractive. The benzo[*d*]imidazole ring is considered to be a privileged scaffold in drug discovery and has demonstrated its versatility in the development of molecules with varied pharmacologic properties. On the other hand, the main psychoactive component of *Cannabis sativa*, delta-9-tetrahydrocannabinol (THC), can be structurally described as an aliphatic terpenoid motif fused to an aromatic polyphenolic (resorcinol) structure. Inspired by the structure of this phytocannabinoid, we combined different natural product motifs with a benzo[*d*]imidazole scaffold to obtain a new library of compounds targeting the CB2 receptor. Here, we synthesized 26 new compounds, out of which 15 presented CB2 binding and 3 showed potent agonist activity. SAR analysis indicated that the presence of bulky aliphatic or aromatic natural product motifs at position 2 of the benzo[*d*]imidazoles ring linked by an electronegative atom is essential for receptor recognition, while substituents with moderate bulkiness at position 1 of the heterocyclic core also participate in receptor recognition. Compounds **5**, **6**, and **16** were further characterized through in vitro cAMP functional assay, showing potent EC50 values between 20 and 3 nM, and compound **6** presented a significant difference between the EC50 of pharmacologic activity (3.36 nM) and IC50 of toxicity (30–38 µM).

## 1. Introduction

Natural products are known as a major source of chemical diversity, providing medicinal products throughout history [1,2]. Notable examples where natural products have become successful therapeutic agents include the analgesic morphine, the anticancer drug vincristine, and the antimalarial artemisinin [2].

Evidence, thus far, shows the ubiquitous presence of the components of the endocannabinoid system (ECS) across the human body, including both central and peripheral tissues. The widespread nature of the ECS highlights its role as a broad-spectrum modulator of homeostasis, bringing forth its therapeutic potential for the treatment of inflammation, pain management, cardiovascular regulation, metabolic disorders, cancer, and neurodegenerative disorders [3,4,5].

The ECS is a lipid signaling system, and the primary receptor proteins are the cannabinoid receptors type I (CB1) and type II (CB2), which are members of the G-protein-coupled receptor (GPCR) family and signal through G_i_-mediated mechanisms. Endogenous ligands that activate the cannabinoid receptors include 2-AG (2-arachidonoyl glycerol) and anandamide (AEA [*N*-arachidonoyl ethanolamine]), although other structurally related lipids have also been identified as endocannabinoids. Additionally, enzymes associated with the biosynthesis of endocannabinoids include NAPE-PLD (*N*-acylphosphatidylethanolamine-specific phospholipase D-like hydrolase) and DAGLα/β (Diacylglycerol lipase α/β), which catalyze the biosynthesis of AEA and 2-AG, respectively, as well as those responsible for endocannabinoid degradation, such as FAAH (fatty acid amide hydrolase) and MAGL (monoacylglycerol lipase) [6,7].

Both the CB1 and CB2 receptors mediate anti-inflammatory responses but show different patterns of expression [8]. While the CB1 receptor is mainly expressed within the CNS where it can be associated with the psychoactive property of marijuana, CB2 is most abundant in the immune tissues. Thus, the development of CB2-selective ligands opens an opportunity to regulate inflammatory responses while avoiding the psychoactive effects associated with CB1 activation [9,10].

Within this context, our group has previously worked on the development of benzo[*d*]imidazole-based small molecules that target the cannabinoid receptors [11,12,13,14,15,16]. This heterocycle can be considered as a privileged scaffold in drug discovery and has demonstrated its versatility as a framework to develop molecules with diverse pharmacologic properties [17,18,19,20,21]. On the other hand, tetrahydrocannabinol (THC) is the main psychoactive component of *Cannabis sativa*, with affinity for both cannabinoid receptors (Figure 1) [22,23]. This molecule can be structurally related to a terpenoid motif fused to an aromatic polyphenolic (resorcinol) structure. Inspired by the chemical structure of this phytocannabinoid and considering our previous experience with benzo[*d*]imidazole derivatives as an effective scaffold to develop cannabinoid ligands, we sought to combine different natural product motifs with the heterocyclic core to access a new library of compounds targeting the cannabinoid receptors. Various medicinal properties, such as antioxidant, antibacterial, and anti-inflammatory action, have been commonly associated with natural products [24,25,26], with many of them being designated as GRAS (Generally Recognized as Safe) substances by the FDA [27]. Therefore, the strategy to combine natural products with a synthetic scaffold can complement the chemical space of cannabinoid ligands with interesting pharmacologic properties.

Herein, we report the synthesis and pharmacological characterization of a new series of CB2 ligands based on natural product motifs conjugated to a benzo[*d*]imidazole. The compounds were evaluated for CB2 affinity using a radioligand binding assay and agonist activity through cAMP accumulation assay.

## 2. Results

### 2.1. Design of Compounds

The general structure of the synthesized compounds is outlined in Table 1. Our previous studies on benzo[*d*]imidazole derivatives suggested that substitutions at position 2 of the heterocycle with bulky and hydrophobic groups were preferred for CB2 affinity, and the presence of electronegative substituents at the same position could be favorable [16]. Therefore, hydrocarbons, such as adamantane, terpenes, and polyphenols (resorcinol), which fulfill these characteristics, were chosen to be functionalized at position 2 of the benzo[*d*]imidazole scaffold using either an oxygen or sulfur linker. Additionally, to probe the steric requirements at position 1 of the heterocycle, both short-chain hydrocarbon and bulky aromatic groups were substituted at this position.

### 2.2. Chemistry

2-alkoxybenzo[*d*]imidazoles were synthesized by reacting the corresponding alcohol reagent in an aromatic nucleophilic substitution reaction. For aliphatic alcohol derivatives (Figure 1), 2-chlorobenzo[*d*]imidazole was first alkylated at position 1 with the corresponding alkyl halides to obtain 1-alkyl-2-chlorobenzo[*d*]imidazoles **I**–**III**. Then, alcohols *l*-menthol, 1-adamantanol, and geraniol were reacted with **I**–**III** in the presence of NaH through nucleophilic aromatic substitution to yield products **4**–**6**, **10**–**12**, and **16**–**17**. Unfortunately, naphthyl derivative of 1-adamantanol could not be obtained through this procedure, possibly due to steric hindrance of the bulky napthyl substituent, which impedes the substitution. Additionally, when the same synthetic methodology was carried out using anisyl alcohol, the alcoxy-substituted product could not be identified, and only a side product presumed to be 1-alkyl-2-benzo[*d*]imidazolone (analyzed by NMR) was identified.

For aryl alcohol derivatives, 1-alkyl-2-benzylsulphonylbenzimidazoles **IV**–**VI** were first synthesized, as described in Figure 2. 2-(benzylthio)-1H-benzo[*d*]imidazole was first alkylated at position 1 with the corresponding alkyl halide, and the resulting dialkylated thioether was oxidized to the corresponding sulphone derivative using *m*-CPBA. Lastly, sulphones **VII**–**IX** were reacted with resorcinol and 1,1-dimethylheptylresorcinol via nucleophilic aromatic substitution reaction to yield products **21**–**26**.

The synthesis of 2-thioxybenzo[*d*]imidazole derivatives is described in Figure 3 and Figure 4. Compounds **1**–**3** were obtained by first tosylating *l*-menthol and geraniol using the procedure described by Hartung et al. [28]. The obtained products were then employed in the alkylation of 2-mercaptobenzo[*d*]imidazole to yield compounds **XII**–**XIII**, which were alkylated with the corresponding alkyl halide using the same procedure described before to obtain compounds **1**–**3** and **7**–**9**.

For compounds **13**–**15**, 1-adamantanol was reacted with 2-mercaptobenzo[*d*]imidazole via S_N_1 conditions using CF_3_COOH as a solvent, and the obtained thioxybenzo[*d*]imidazole **XIV** was alkylated with the corresponding naphthyl, benzyl, and ethyl halides at position 1, as described above. For compounds **18**–**20**, 2-mercaptobenzo[*d*]imidazole was selectively monoalkylated at position 2 using an equivalent of anisyl chloride in the presence of TBAB and a base to give compound **XV**, which was further alkylated at position 1 using the same alkyl halides mentioned above.

### 2.3. Radioligand Displacement Assay

To assess ligand binding to CB2 receptors, radioligand displacement assay at a single dose (10 μM) was performed in membranes obtained from recombinant CHO cells expressing human CB2 receptors (Eurofins Cerep SA, France). The results are presented in Table 2 and Figure 2. Out of 26 compounds, more than 50% of the molecules presented >50% displacement, while 20% of the compounds (**5**, **6**, **16**, **19**, **22**) presented >80% displacement of radioligand binding at a 10 μM dose. Compounds **5**, **6**, **16**, **19**, and **22** were further tested for CB2 receptor activation (agonist activity, see below) and CB2/CB1 selectivity.

### 2.4. cAMP Accumulation Assay

Compounds were further characterized through in vitro functional assays by measuring the variation in forskolin-induced cAMP accumulation (Eurofins Cerep services). The tested compounds diminished the accumulation of cyclic AMP, indicating activity as agonists (Figure 3).

As shown in Table 3, all the values of EC50 varied within the nanomolar range of activity, with compound **16** presenting an EC50 value of 20 nM, compound **5** 14 nM, and the most potent, compound **6** 3.36 nM.

### 2.5. CB1/CB2 Receptor Selectivity

To assess the compound selectivity between CB1 and CB2 receptors, binding constants were determined through a radioligand displacement assay by testing concentration–response curves for compounds **5**, **6**, **16**, **19**, and **22**. As shown in Figure 4 and summarized in Table 4, three of the tested compounds (**5**, **19**, and **22**) presented moderate binding affinity in a low micromolar range to both CB1 and CB2 receptors. However, compounds **6** and **16** showed improved selectivity profiles, with at least ten-fold higher affinity toward the CB2 receptor. Noteworthily, although the tested compounds showed moderate binding affinities within the micromolar range, agonist activity measured through functional assays presented nanomolar values, with compound **6** being the most potent agonist (EC50 = 3.36 nM).

### 2.6. Molecular Docking

Compounds **5**, **6**, and **16** were further studied through molecular docking to gain insight into their binding mode within the orthosteric pocket of the CB2 receptor (Figure 5A–D). Docking was performed using the available cryo-EM structure of the CB2 receptor bound to WIN55212-2 (PDB ID: 6PT0).

### 2.7. Neutral Red Uptake Assay

Compound **6** was tested through a neutral red uptake assay, and cell viability was measured. Neutral red consists of a cationic dye that accumulates in lysosomes. Uptake of neutral red depends on a viable cell’s capacity to maintain acidic pH in the interior of lysosomes [29]. Figure 6 and Table 5 shows the data for cell viability in two different cell lines: HEK-293 (Human Embryonic Kidney 293; non-cancerous cell lines of renal tissue) and MCF-7 (Michigan Cancer Foundation-7, breast cancer cell line). The results indicate that an IC50 of viability for compound **6** presents a value of 30 µM in HEK-293 cells and 38 µM in MCF-7 cells.

## 3. Discussion

The structural information obtained from Table 2 and Figure 2 showed that derivatives with either aliphatic or aromatic natural product motifs presented activities that spanned from limited to excellent, suggesting that the chemical nature of the motif has little impact on CB2 receptor recognition. Additionally, bulky groups such as adamantyl were well tolerated, but longer motifs in R2 seem to be detrimental for affinity, as geranyl and DMH derivatives presented lower percentages of radioligand displacement (compounds **10**–**12** and **24**–**26**). Regarding the effect of the linker atom, by comparing compounds **1**–**3** (sulfur linker) with compounds **4**–**6** (oxygen linker), higher inhibition was observed for oxygen linker derivatives. This is also true when comparing compounds **14**–**15** (sulfur linker) and **16**–**17** (oxygen linker) with oxygen derivatives presenting equivalent or superior inhibition percentage.

Therefore, the data indicate that the presence of an oxygen linker is more favorable for affinity, in agreement with our previous QSAR study [16], which suggested that the presence of electronegative atoms at position 2 of benzo[*d*]imidazoles could increase the activity. Some exceptions are geranyl derivatives (compounds **7**–**12**), where alkoxy derivatives present lower binding inhibition than their thioxy counterparts. Nevertheless, these geranyl derivatives can be considered as part of the “elongated” series of compounds, which were unfavorable for activity, as discussed before. Thus, maintaining the adequate size of the substituent at position 2 of benzo[*d*]imidazoles seems to be of greater importance than the presence of electronegative atoms at the same position. Additionally, the difference in atomic size between oxygen and sulphur linker atoms, which determines a change in the angle between the two substituent groups, could play a role in the proper orientation of the compounds within the binding site.

Regarding the effect of R1 substituents, again, the size of the introduced group affects binding affinity. The presence of an ethyl group at R1 yielded compounds with moderate to excellent affinity (compounds **3**, **6**, **9**, **15**, **17**, **20**, **23**, and **26**). Nevertheless, changing this group to a benzyl substituent maintained or even enhanced activity (compounds **2**, **5**, **8**, **14**, **16**, **19**, **22**). However, the introduction of a bulkier naphthalen-1-ylmethyl substituent greatly diminished receptor recognition (compounds **4**, **7**, **13**, **18** and **21**) and, in some cases, was detrimental for binding (compound **1**). Thus, the data suggest that the presence of lipophilic groups with moderate bulkiness is preferable on R1.

Based on the results obtained from the radioligand displacement assay, compounds **5**, **6**, **16**, **19**, and **22** were selected and analyzed through a cAMP accumulation assay in recombinant cells expressing CB2 receptors, and agonist activity was confirmed (Table 3). Nevertheless, statistical analyses could not be performed for compounds **19** and **22** (EC50 not determined). In the case of compounds **5**, **6**, and **16**, agonist activities in the nanomolar range were observed, with compound **6** (a menthol derivative) presenting the best profile, with an EC50 of 3.3 nM. Interestingly, although this compound presented potent agonist behavior over the CB2 receptor (as measured by cAMP accumulation), the affinity was near the micromolar range, indicating that compound **6** exerts a high pharmacological response at moderate affinity. Regarding the binding selectivity between CB1 and CB2 receptors (Table 4), the five tested compounds showed moderate selectivity indexes, but compound **6**, the most potent identified compound, presented 20-fold higher recognition toward the CB2 receptor according to the Ki(CB1)/Ki(CB2) ratio, turning it as a potent and selective derivative for the CB2 receptor over CB1.

The analysis of the binding modes of compounds **5**, **6**, and **16** compared to that of WIN55212-2 (PDB ID: 6PT0) showed that all compounds adopt similar binding modes to the agonist WIN55212-2 (Figure 5A), maintaining most of the described interactions in the orthosteric site. As the indole ring, the benzo[*d*]imidazole heterocycle acts as a central core directing the substituent groups at positions 1 and 2 towards TM2 and TM5, respectively. This binding mode produces a superposition of the terpenoid motif at position 2 with the naphthalene ring of WIN55212-2, while substituents at position 1 coincide with the morpholine moiety of the agonist.

Figure 5B–D present the binding interactions established by compounds **5**, **6**, and **16** within the orthosteric pocket of the receptor. The predominance of hydrophobic interactions is seen in accordance with the highly lipophilic nature of the CB2 receptor binding pocket. Two hydrophobic pockets can be identified within the binding site. One of them extends into TM2 and the other one is composed of TM5 residues and capped with aromatic residues of ECL2. The obtained docking poses show that the natural motifs of compounds **5**, **6**, and **16** extend towards the pocket in TM2, engaging in hydrophobic interactions with Phe87, Phe91, and/or Phe94, while the second pocket harbors the substituents at position 1 also through hydrophobic and pi-stacking interactions with Ile110, Val113, Phe183, Ile186, and Trp194 (Figure 5B–D). In this way, the heterocyclic core acts as a bridging scaffold between these two pockets and, at the same time, forms hydrophobic contacts with one of the toggle-switch residues Phe117, important for receptor activation. Additionally, comparison of the docking poses in Figure 5 shows that the most potent compound **6** can directly interact with residue Ser285, which has been described to play a role in ligand efficiency [30].

Lastly, the toxicity of compound **6** was evaluated in vitro through the neutral red uptake assay. Experiments were performed in both cancerous (MCF-7) and non-cancerous (HEK-293) cell line models, showing that compound **6** decreased cell viability to 50% at 38 µM and 30 µM, respectively. The results showed little difference in toxicity between cancerous and non-cancerous cell lines. Nevertheless, considering that compound **6** presents an EC50 of 3.3 nM in the cAMP accumulation assay, there is a difference by five orders of magnitude between pharmacologic activity and toxicity. Based on the obtained results, compound **6** represents a safe, potent, and selective derivative for CB2 receptor.

In summary, new potent and selective CB2 ligands based on natural product motifs linked to a benzo[*d*]imidazole core were obtained. SAR analysis suggested that the presence of bulky aliphatic or aromatic natural product motifs at position 2 of the benzo[*d*]imidazole ring is essential for receptor recognition, linked preferably by an electronegative atom. Furthermore, the presence of substituents with moderate bulkiness at position 1 of the heterocyclic scaffold is also important for receptor recognition, with a benzyl group being the optimal substituent. Functional evaluation identified five compounds with agonist activity for the CB2 receptor. Docking studies support a common binding mode for the analyzed compounds. The high potency to inhibit cAMP accumulation, albeit having moderate affinities over the CB2 receptor, highlights the importance of complementing both binding and functional data as well as showing that great affinity is not needed to perform a potent pharmacological response. Finally, the cell viability assay showed a low toxicity profile for the most potent compound. Future evaluation through different assays will be useful to further characterize the pharmacological profile of the new ligands.

## 4. Materials and Methods

### 4.1. Chemistry

Reagents were purchased from commercial suppliers and used without further purification. Anhydrous solvents were prepared by storing over activated molecular sieves (pore size 3–4 Å) for at least 2 days. The sieves were previously activated by heating in an oven at 300 °C. Reactions were monitored via thin-layer chromatography using precoated aluminum plates (Merck TLC Silica gel 60 F254). Spots were visualized using UV light (254 nm and 366 nm), iodine chamber, Dragendorff’s reagent (reveal basic nitrogen), or *p*-anisaldehyde solution (reveal terpenes). Column chromatography was carried out using Merck silica gel 60 (230–400 mesh). Plates for preparative thin-layer chromatography were prepared in glass sheets (dimensions 20 × 20) using Merck silica gel PF254 containing gypsum. Measurements of NMR spectra were performed on a Bruker Advance 400 (1H NMR: 400 MHz; 13C NMR: 101 MHz). Chemical shifts are reported in parts per million (ppm) relative to chloroform-d (CDCl3; δ7.26), dimethylsulfoxide-d6 (DMSO; δ2.50), or acetone-d6 (CD3COCD3; δ2.05). Coupling constants (J) are expressed in hertz (Hz). Melting points were determined on a Stuart SMP10 apparatus (see Appendix A).

#### 4.1.1. General Procedure for the Synthesis of Compounds **I**–**III**

Here, 1 equivalent of 2-chlorobenzimidazol and 1.2 equivalent of NaH (as 60% oil disp.) were stirred at room temperature for 30 min under N_2_ atmosphere and dry AcCN as solvent. Then, 1 equivalent of alkyl halide was added dropwise, and the reaction was heated in an oil bath at 40 °C overnight. Excess NaH was inactivated with MeOH, and the suspension was filtered and washed with DCM. The organic phase was distilled under vacuum, obtaining an oily residue that solidified over time.

*2-chloro-1-(naphthalen-1-ylmethyl)-1H-benzo[d]imidazole (***I***)*. **Yield:** 98%. White solid. **^1^H NMR** (400 MHz, Chloroform-*d*) δ 8.16 (d, *J* = 8.4 Hz, 1H), 8.06 (dd, *J* = 8.1, 1.5 Hz, 1H), 7.96–7.90 (m, 1H), 7.81–7.68 (m, 1H), 7.47–7.38 (m, 1H), 7.32 (ddd, *J* = 8.3, 7.2, 1.1 Hz, 1H), 7.22 (d, *J* = 8.1 Hz, 1H), 6.80 (dd, *J* = 7.2, 1.3 Hz, 1H), 5.96 (s, 1H). **^13^C NMR** (101 MHz, CDCl_3_) δ 141.34, 141.16, 135.26, 133.73, 130.27, 129.88, 129.21, 128.67, 126.95, 126.31, 125.49, 123.70, 123.24, 123.21, 121.99, 119.44, 110.15, 45.76. Purified by column chromatography using DCM:AcOEt (4:1).*1-benzyl-2-chloro-1H-benzo[d]imidazole (***II***).***
Yield:
**
92%. White solid. **^1^H NMR** (400 MHz, Chloroform-*d*) δ 7.70–7.63 (m, 1H), 7.28–7.18 (m, 4H), 7.18–7.14 (m, 2H), 7.13–7.07 (m, 2H), 5.28 (s, 2H). **^13^C NMR** (101 MHz, CDCl_3_) δ 141.85, 140.80, 135.18, 135.06, 129.02, 128.20, 126.81, 123.39, 122.87, 119.53, 109.92, 47.90. Recrystallized in H_2_O:EtOH.*2-chloro-1-ethyl-1H-benzo[d]imidazole (***III***)*. **Yield:** 97%. White solid. **^1^H NMR** (400 MHz, DMSO-*d*_6_) δ 7.60 (d, *J* = 8.0 Hz, 2H), 7.38–7.16 (m, 2H), 4.27 (q, *J* = 7.2 Hz, 2H), 1.30 (t, *J* = 7.2 Hz, 3H). **^13^C NMR** (101 MHz, DMSO) δ 141.69, 139.83, 134.97, 123.35, 122.78, 119.09, 110.81, 39.43, 14.89. Purified by column chromatography using DCM:AcOEt (4:1).

#### 4.1.2. General Procedure for the Synthesis of Compounds **IV**–**VI**

The synthetic procedure was adapted from Rao et al. [31]. In brief, 1 mmol of 2-(benzylthio)-1H-benzo[*d*]imidazole, 4 mmol of K_2_CO_3_, tetrabutylammonium bromide (TBAB), and 1 mmol of the corresponding alkyl halide were suspended in DMF, and the mixture was stirred overnight. The mixture was poured over water, and the aqueous phase was extracted with DCM and AcOEt. The combined organic phase was dried over Na_2_SO_4_ and the solvent removed in vacuo. Products were purified using column chromatography.

*2-(benzylthio)-1-(naphthalen-1-ylmethyl)-1H-benzo[d]imidazole (***IV***)*. **Yield:** 70%. Beige solid. **^1^H NMR** (400 MHz, Chloroform-*d*) δ 7.91 (d, *J* = 7.9 Hz, 1H), 7.86 (d, *J* = 8.0 Hz, 2H), 7.72 (d, *J* = 8.4 Hz, 1H), 7.58–7.46 (m, 2H), 7.45–7.34 (m, 2H), 7.31–7.15 (m, 5H), 7.09 (t, *J* = 7.7 Hz, 1H), 6.97 (d, *J* = 8.1 Hz, 1H), 6.61 (d, *J* = 7.2 Hz, 1H), 5.60 (s, 2H), 4.65 (s, 2H). **^13^C NMR** (101 MHz, CDCl_3_) δ 152.24, 143.84, 136.84, 136.57, 133.73, 130.65, 130.46, 129.21, 129.15, 128.74, 128.37, 127.77, 126.76, 126.18, 125.60, 123.39, 122.44, 122.33, 122.24, 118.67, 109.55, 45.33, 37.50. Purified via column chromatography using Hexane:AcOEt (6:1), then recrystallized in EtOH:AcOEt.*1-benzyl-2-(benzylthio)-1H-benzo[d]imidazole (***V***)*. **Yield:** 83%. Beige solid. **^1^H NMR** (400 MHz, Chloroform-*d*) δ 7.77 (d, *J* = 8.0 Hz, 1H), 7.42 (dd, *J* = 7.8, 1.8 Hz, 2H), 7.34–7.22 (m, 7H), 7.17 (d, *J* = 4.1 Hz, 2H), 7.10 (dd, *J* = 7.2, 2.4 Hz, 2H), 5.23 (s, 2H), 4.65 (s, 2H). **^13^C NMR** (101 MHz, CDCl_3_) δ 151.77, 143.76, 136.77, 136.28, 135.70, 129.20, 128.90, 128.75, 127.96, 127.75, 126.94, 122.29, 122.16, 118.57, 109.38, 47.64, 37.58. Purified via column chromatography using Hexane:AcOEt (3:1), then recrystallized in EtOH:H_2_O.*2-(benzylthio)-1-ethyl-1H-benzo[d]imidazole (***VI***)*. **Yield:** 85%. Yellow oil. **^1^H NMR** (400 MHz, Chloroform-*d*) δ 7.76–7.68 (m, 1H), 7.43–7.37 (m, 2H), 7.32–7.18 (m, 6H), 4.62 (s, 2H), 4.06 (q, *J* = 7.3 Hz, 2H), 1.30 (t, *J* = 7.3 Hz, 4H). **^13^C NMR** (101 MHz, CDCl_3_) δ 151.37, 144.08, 137.17, 136.06, 129.49, 129.08, 128.06, 122.31, 122.19, 118.81, 109.07, 39.27, 37.59, 14.93. Purified via column chromatography using a gradient from DCM to DCM:MeOH 5%.

#### 4.1.3. General Procedure for the Synthesis of Compounds **VII**–**IX**

Here, 1 equivalent of compound **IV**–**VI** was dissolved in DCM, and the solution was cooled using an ice bath. Further, 2 equivalents of m-CPBA were carefully added to the agitating solution, and the mixture was gradually heated to room temperature and stirred overnight. The resulting suspension was filtered, and the organic layer concentrated, recovering a solid, which was resuspended in a saturated solution of NaHCO_3_ and then filtered. For oily residues, the crude reaction was extracted with a solution of NaHCO_3_; the organic layer was dried with Na_2_SO_4_ and then distilled under vacuum.

*2-(benzylsulfonyl)-1-(naphthalen-1-ylmethyl)-1H-benzo[d]imidazole (***VII***)*. **Yield:** 88%. Beige solid. **^1^H NMR** (400 MHz, Chloroform-*d*) δ 7.94 (d, *J* = 8.3, 1.0 Hz, 1H), 7.80 (t, *J* = 8.0, 1.6 Hz, 2H), 7.66 (d, *J* = 8.3 Hz, 1H), 7.49 (qd, *J* = 14.8, 8.3, 6.9, 1.5 Hz, 2H), 7.37–7.31 (m, 2H), 7.28–7.19 (m, 3H), 7.18–7.13 (m, 2H), 7.08 (t, *J* = 8.2, 7.2 Hz, 1H), 6.97 (d, *J* = 8.3, 1.0 Hz, 1H), 6.29 (dd, *J* = 7.2, 1.2 Hz, 1H), 5.76 (s, 2H), 4.74 (s, 2H). **^13^C NMR** (101 MHz, CDCl_3_) δ 147.07, 141.27, 135.74, 133.55, 131.48, 130.58, 130.05, 129.30, 129.03, 128.87, 128.26, 126.73, 126.52, 126.30, 126.10, 125.45, 124.38, 122.86, 122.00, 121.95, 111.57, 61.70, 46.09. Recrystallized in EtOH:DCM*1-benzyl-2-(benzylsulfonyl)-1H-benzo[d]imidazole (***VIII***)*. **Yield:** 82%. Yellow solid. **^1^H NMR** (400 MHz, Chloroform-*d*) δ 7.97–7.93 (m, 1H), 7.43–7.34 (m, 3H), 7.32–7.19 (m, 8H), 7.06–6.97 (m, 2H), 5.41 (s, 2H), 4.77 (s, 2H). **^13^C NMR** (101 MHz, CDCl_3_) δ 146.64, 141.26, 135.53, 135.26, 131.43, 129.27, 128.86, 128.85, 128.08, 126.95, 126.41, 126.20, 124.27, 121.92, 111.57, 61.62, 48.51. Recrystallized in Hexane:AcOEt.*2-(benzylsulfonyl)-1-ethyl-1H-benzo[d]imidazole (***IX***)*. **Yield:** 93%. Yellow solid. **^1^H NMR** (400 MHz, Chloroform-*d*) δ 7.90 (d, *J* = 7.9 Hz, 1H), 7.53–6.98 (m, 8H), 4.81 (s, 2H), 4.17 (q, *J* = 7.1 Hz, 2H), 1.19 (t, *J* = 7.2 Hz, 3H). **^13^C NMR** (101 MHz, CDCl_3_) δ 146.26, 141.26, 134.89, 131.30, 129.19, 128.79, 126.56, 125.87, 124.04, 121.94, 110.67, 61.62, 40.22, 15.27. Purified via column chromatography using Hexane:AcOEt (3:1).

#### 4.1.4. General Procedure for the Synthesis of Derivatives **XII**–**XIII** and **XV**

The synthetic procedure was adapted from Rao et al. [31], where 1 mmol of 2-mercaptobenzimidazol, 4 mmol of K_2_CO_3_, tetrabutylammonium bromide (TBAB), and 1 mmol of the corresponding tosylate derivative were dissolved in DMF, and the mixture was heated overnight in an oil bath at 70 °C. The mixture was poured over water, and the aqueous phase was extracted with DCM. The organic phase was dried over Na_2_SO_4_ and the solvent removed in vacuo. For compound **XV**, 4-methoxybenzyl chloride was used instead of a tosylate derivate. Products were purified via recrystallization.

*2-((2-isopropyl-5-methylcyclohexyl)thio)-1H-benzo[d]imidazole (***XII***)*. **Yield:** 46%. White solid. **^1^H NMR** (400 MHz, DMSO-*d*_6_) δ 12.46 (s, 1H), 7.46–7.39 (m, 2H), 7.14–7.06 (m, 2H), 4.53 (dq, *J* = 4.4, 2.7 Hz, 1H), 2.02 (dq, *J* = 13.4, 2.9 Hz, 1H), 1.89–1.70 (m, 3H), 1.57 (dp, *J* = 9.6, 6.6 Hz, 1H), 1.40 (ddd, *J* = 13.7, 11.7, 3.1 Hz, 1H), 1.32–1.21 (m, 1H), 1.09–0.93 (m, 2H), 0.92 (d, *J* = 3.5 Hz, 3H), 0.90 (d, *J* = 3.5 Hz, 3H), 0.85 (d, *J* = 6.5 Hz, 3H). **^13^C NMR** (101 MHz, DMSO) δ 150.80, 121.67, 48.68, 47.89, 41.66, 35.01, 30.78, 27.66, 27.07, 22.37, 21.30, 20.91. Recrystallized in AcOEt:EtOH.*(E)-2-((3,7-dimethylocta-2,6-dien-1-yl)thio)-1H-benzo[d]imidazole (***XIII***)*. **Yield:** 44%. Yellow oil. **^1^****H NMR** (400 MHz, Chloroform-*d*) δ 9.59 (s, 1H), 7.66 (s, 1H), 7.36 (s, 1H), 7.23–7.14 (m, 2H), 5.41 (t, *J* = 8.0 Hz, 1H), 5.05 (t, *J* = 6.5 Hz, 1H), 3.97 (d, *J* = 7.9 Hz, 2H), 2.11–1.98 (m, 4H), 1.67 (s, 3H), 1.66 (s, 3H), 1.58 (s, 3H). **^13^C NMR** (101 MHz, CDCl_3_) δ 150.36, 141.63, 131.92, 123.72, 122.32, 118.37, 39.51, 31.18, 26.31, 25.66, 17.73, 16.19. Purified via column chromatography using Hexane:AcOEt (3:1).*2-((4-methoxybenzyl)thio)-1H-benzo[d]imidazole (***XV***)*. **Yield:** 85%. White solid. **^1^****H NMR** (400 MHz, DMSO-*d*_6_) δ 12.55 (s, 1H), 7.55 (s, 1H), 7.37 (d, *J* = 8.5 Hz, 3H), 7.16–7.09 (m, 2H), 6.86 (d, *J* = 8.6 Hz, 2H), 4.52 (s, 2H), 3.71 (s, 3H). **^13^C NMR** (101 MHz, DMSO) δ 158.54, 149.86, 143.67, 135.43, 130.07, 129.37, 121.62, 121.14, 117.41, 113.88, 110.31, 55.03, 39.52, 34.78. Recrystallized in H_2_O:EtOH.

#### 4.1.5. Synthesis of Compound **XIV**

Here, 1 mmol of 2-mercaptobenzimidazol and 1 mmol of 1-adamatanol were dissolved in 1.33 mL of CF_3_COOH and heated in an oil bath at 80 °C for 1 h. Then, 5 mL of a solution of EtOH:H_2_O (1:1) was added, and the reaction was neutralized with NH_3_(ac). A precipitate formed, which was filtered and recrystallized with H_2_O:EtOH (1:9), obtaining white crystals.

*2-(adamantan-1-ylthio)-1H-benzo[d]imidazole (***XIV***)*. **Yield:** 65%. White solid. **^1^H**
**NMR** (400 MHz, DMSO-*d*_6_) δ 7.66–7.36 (m, 2H), 7.30–6.97 (m, 2H), 1.98 (s, 9H), 1.60 (s, 6H). **^13^C NMR** (101 MHz, DMSO) δ 144.83, 121.94, 50.56, 43.36, 39.52, 35.49, 29.49. Recrystallized in H_2_O:EtOH.

#### 4.1.6. General Procedure for the Synthesis of Derivatives **1**–**3**, **7**–**9**, **13**–**15**, and **18**–**20**

Here, 1 equivalent of 2-thioxybenzimidazol **XII**–**XV**, 4 equivalents of K_2_CO_3_, 0.05 equivalent of tetrabutylammonium bromide (TBAB), and 1 equivalent of the corresponding alkyl halide were dissolved in DMF, and the mixture was stirred overnight at room temperature. The mixture was poured over water, and the resulting precipitate was filtered and washed with water. When a filterable precipitate was not formed, the aqueous phase was extracted with DCM, the organic layer was dried over Na_2_SO_4,_ and the solvent removed in vacuo. Products were purified via column chromatography or preparative plate or recrystallization.

*2-((2-isopropyl-5-methylcyclohexyl)thio)-1-(naphthalen-1-ylmethyl)-1H-benzo[d]imidazole (***1**). **Yield:** 75%. white solid. **^1^H NMR** (400 MHz, Chloroform-*d*) δ 8.33 (d, *J* = 8.3 Hz, 2H), 8.15 (d, *J* = 8.0 Hz, 1H), 8.00 (t, *J* = 8.7 Hz, 2H), 7.82 (dt, *J* = 21.1, 7.1 Hz, 2H), 7.55–7.48 (m, 1H), 7.44 (t, *J* = 7.5 Hz, 1H), 7.35–7.21 (m, 2H), 6.92 (d, *J* = 7.2 Hz, 1H), 6.04 (s, 2H), 4.95 (s, 1H), 2.47 (dq, *J* = 14.0, 3.1 Hz, 1H), 2.08–1.87 (m, 4H), 1.84–1.73 (m, 1H), 1.72–1.62 (m, 1H), 1.54–1.43 (m, 1H), 1.32–1.16 (m, 3H), 1.12 (d, *J* = 6.7 Hz, 3H), 1.08 (d, *J* = 6.4 Hz, 3H). **^13^C NMR** (101 MHz, CDCl_3_) δ 153.31, 143.88, 136.43, 133.70, 130.78, 130.51, 129.07, 128.18, 126.57, 126.01, 125.49, 123.36, 122.28, 121.89, 121.82, 118.34, 109.20, 50.01, 48.28, 45.35, 41.48, 35.10, 30.68, 27.53, 26.89, 22.05, 21.02, 20.83. Purified via preparative plate using Hexane:AcOEt (6:1) and then recrystallized in MeOH.*1-benzyl-2-((2-isopropyl-5-methylcyclohexyl)thio)-1H-benzo[d]imidazole (***2***)*. **Yield:** 39%. Yellow solid. **^1^H NMR** (400 MHz, DMSO-*d*_6_) δ 7.60–7.53 (m, 1H), 7.46–7.40 (m, 1H), 7.32–7.28 (m, 2H), 7.26 (d, *J* = 7.1 Hz, 1H), 7.18–7.12 (m, 4H), 5.41 (d, *J* = 2.2 Hz, 2H), 2.08–1.96 (m, 1H), 1.83 (d, *J* = 12.2 Hz, 1H), 1.75–1.67 (m, 2H), 1.54–1.47 (m, 1H), 1.44–1.36 (m, 1H), 1.33–1.25 (m, 1H), 1.08–0.91 (m, 2H), 0.89 (dd, *J* = 2.8, 6.6 Hz, 6H), 0.82 (d, *J* = 6.3 Hz, 3H), 0.76 (dd, *J* = 6.4, 10.6 Hz, 1H). **^13^C NMR** (101 MHz, DMSO) δ 152.30, 143.58, 136.91, 136.48, 129.08, 128.03, 127.28, 122.09, 122.06, 118.11, 110.25, 50.26, 47.81, 47.03, 41.45, 34.92, 30.81, 27.74, 27.16, 22.33, 21.26, 20.83. Purified via preparative plate using Hexane:AcOEt (6:1).*1-ethyl-2-((2-isopropyl-5-methylcyclohexyl)thio)-1H-benzo[d]imidazole (***3***)*. **Yield:** 68%. Yellow oil. **^1^H NMR** (400 MHz, Chloroform-*d*) δ 7.77–7.69 (m, 1H), 7.33–7.26 (m, 1H), 7.26–7.17 (m, 2H), 4.78–4.65 (m, 1H), 4.20 (q, *J* = 7.3 Hz, 2H), 2.32–2.19 (m, 1H), 2.01–1.89 (m, 2H), 1.88–1.78 (m, 1H), 1.76–1.60 (m, 1H), 1.53–1.44 (m, 1H), 1.43 (t, *J* = 7.7, 7.3 Hz, 3H), 1.25–1.12 (m, 2H), 1.09–1.01 (m, 1H), 0.99 (t, *J* = 5.9 Hz, 6H), 0.92 (d, *J* = 6.5 Hz, 3H). **^13^C NMR** (101 MHz, CDCl_3_) δ 152.11, 143.91, 135.81, 121.59, 121.54, 118.33, 108.46, 77.16, 49.56, 48.43, 41.63, 38.85, 35.32, 30.88, 27.77, 27.16, 22.18, 21.18, 20.96, 14.67. Purified via preparative plate using Hexane:AcOEt (20:1).*(E)-2-((3,7-dimethylocta-2,6-dien-1-yl)thio)-1-(naphthalen-1-ylmethyl)-1H-benzo[d]imidazole (***7***)*. **Yield:** 54%. Yellow oil. **^1^H NMR** (400 MHz, Chloroform-*d*) δ 8.23 (d, *J* = 8.3 Hz, 1H), 8.09 (d, *J* = 7.8 Hz, 1H), 7.95 (d, *J* = 8.2 Hz, 2H), 7.83–7.69 (m, 2H), 7.49–7.36 (m, 2H), 7.28 (t, *J* = 7.6 Hz, 1H), 7.22 (d, *J* = 7.9 Hz, 1H), 6.87 (dd, *J* = 7.2, 1.2 Hz, 1H), 5.96 (s, 2H), 5.62–5.52 (m, 1H), 5.26–5.17 (m, 1H), 4.26 (d, *J* = 7.9 Hz, 2H), 2.28–2.13 (m, 4H), 1.88 (s, 3H), 1.81 (s, 3H), 1.74 (s, 3H). **^13^C NMR** (101 MHz, CDCl_3_) δ 152.97, 143.80, 141.89, 136.45, 133.70, 131.76, 130.67, 130.45, 129.09, 128.29, 126.66, 126.07, 125.52, 123.78, 123.34, 122.17, 122.14, 122.09, 118.43, 117.79, 109.29, 45.38, 39.57, 31.31, 26.39, 25.67, 17.71, 16.29. Purified via preparative plate using Hexane:AcOEt (6:1).*(E)-1-benzyl-2-((3,7-dimethylocta-2,6-dien-1-yl)thio)-1H-benzo[d]imidazole (***8***)*. **Yield:** 50%. Yellow oil. **^1^H NMR** (400 MHz, Chloroform-*d*) δ 7.77 (d, *J* = 7.9 Hz, 1H), 7.41–7.30 (m, 3H), 7.32–7.18 (m, 5H), 5.48 (t, *J* = 7.9 Hz, 1H), 5.36 (s, 2H), 5.12 (t, *J* = 6.5 Hz, 1H), 4.13 (d, *J* = 7.9 Hz, 2H), 2.21–2.03 (m, 4H), 1.79 (s, 3H), 1.73 (s, 3H), 1.65 (s, 3H). **^13^C NMR** (101 MHz, CDCl_3_) δ 152.45, 143.72, 141.83, 136.16, 135.77, 131.76, 128.83, 127.88, 126.90, 123.78, 122.03, 121.96, 118.34, 117.84, 109.15, 47.62, 39.57, 31.36, 26.37, 25.68, 17.72, 16.28. Purified via preparative plate using Hexane:AcOEt (6:1).*(E)-2-((3,7-dimethylocta-2,6-dien-1-yl)thio)-1-ethyl-1H-benzo[d]imidazole (***9***)*. **Yield:** 68%. Yellow oil. **^1^H NMR** (400 MHz, Chloroform-*d*) δ 7.31 (d, *J* = 7.8 Hz, 1H), 7.07–7.01 (m, 1H), 6.97–6.90 (m, 2H), 6.31 (dd, *J* = 10.8, 17.6 Hz, 1H), 5.33–5.15 (m, 2H), 5.04 (t, *J* = 7.2 Hz, 1H), 3.90 (q, *J* = 7.1 Hz, 2H), 2.32–2.14 (m, 2H), 2.13–2.00 (m, 1H), 1.90–1.77 (m, 4H), 1.59 (s, 3H), 1.45 (s, 3H), 1.31 (t, *J* = 7.1 Hz, 3H). **^13^C NMR** (101 MHz, CDCl_3_) δ 144.16, 131.80, 129.47, 123.66, 120.66, 120.06, 112.95, 112.47, 106.94, 63.36, 38.24, 35.45, 25.62, 25.15, 22.84, 17.47, 13.44. Purified via preparative plate using Hexane:AcOEt (6:1).*2-(adamantan-1-ylthio)-1-(naphthalen-1-ylmethyl)-1H-benzo[d]imidazole (***13***)*. **Yield:** 58%. White solid. **^1^H NMR** (400 MHz, Chloroform-*d*) δ 8.26 (d, *J* = 8.4 Hz, 1H), 8.04 (dd, *J* = 8.1, 18.7 Hz, 2H), 7.90 (d, *J* = 8.4 Hz, 1H), 7.74 (dt, *J* = 7.4, 26.0 Hz, 2H), 7.45–7.37 (m, 2H), 7.37–7.26 (m, 1H), 7.22 (d, *J* = 8.2 Hz, 1H), 6.69 (d, *J* = 7.1 Hz, 1H), 6.17 (s, 2H), 2.24 (s, 6H), 2.19 (s, 3H), 1.81 (s, 6H). **^13^C NMR** (101 MHz, CDCl_3_) δ 148.13, 143.72, 135.67, 133.65, 131.30, 130.34, 129.08, 128.09, 126.59, 126.03, 125.46, 123.26, 123.01, 122.35, 122.27, 119.83, 110.36, 52.87, 45.81, 43.92, 36.00, 30.25. Recrystallized in EtOH:AcOEt.*2-(adamantan-1-ylthio)-1-benzyl-1H-benzo[d]imidazole (***14***)*. **Yield:** 92%. White solid. **^1^H NMR** (400 MHz, Chloroform-*d*) δ 7.88 (d, *J* = 7.9 Hz, 1H), 7.37–7.28 (m, 4H), 7.25 (d, *J* = 4.2 Hz, 2H), 7.18 (d, *J* = 6.5 Hz, 2H), 5.61 (s, 2H), 2.20–2.09 (m, 9H), 1.73 (s, 6H). **^13^C NMR** (101 MHz, CDCl_3_) δ 147.58, 143.64, 136.31, 135.40, 128.75, 127.72, 126.75, 122.92, 122.25, 119.78, 110.29, 52.94, 48.11, 43.92, 36.00, 30.25. Recrystallized in H_2_O:EtOH.*2-(adamantan-1-ylthio)-1-ethyl-1H-benzo[d]imidazole (***15***)*. **Yield:** 77%. White solid. ^1^H NMR (200 MHz, DMSO-*d_6_*) δ 7.64 (d, *J* = 6.9 Hz, 1H), 7.55 (d, *J* = 8.5 Hz, 1H), 7.33–7.13 (m, 2H), 4.32 (q, *J* = 7.2 Hz, 2H), 2.16–1.91 (m, 9H), 1.63 (s, 6H), 1.27 (t, *J* = 7.2 Hz, 3H). **^13^C NMR** (50 MHz, DMSO) δ 146.21, 143.07, 134.56, 122.32, 121.70, 118.66, 110.19, 51.75, 43.12, 35.48, 29.52, 14.92. Recrystallized in H_2_O:EtOH.*2-((4-methoxybenzyl)thio)-1-(naphthalen-1-ylmethyl)-1H-benzo[d]imidazole (***18***)*. **Yield:** 83%. White solid. **^1^H NMR** (400 MHz, Chloroform-*d*) δ 7.95 (d, *J* = 8.1 Hz, 1H), 7.87 (d, *J* = 7.3 Hz, 1H), 7.80 (d, *J* = 8.0 Hz, 1H), 7.72 (d, *J* = 8.4 Hz, 1H), 7.58–7.46 (m, 2H), 7.27 (d, *J* = 8.6 Hz, 2H), 7.21 (dd, *J* = 7.8, 16.0 Hz, 2H), 7.09 (t, *J* = 7.7 Hz, 1H), 7.00 (d, *J* = 8.0 Hz, 1H), 6.77 (d, *J* = 8.2 Hz, 2H), 6.58 (d, *J* = 7.2 Hz, 1H), 5.66 (s, 2H), 4.58 (s, 2H), 3.72 (s, 3H). **^13^C NMR** (101 MHz, CDCl_3_) δ 159.15, 152.32, 143.77, 136.44, 133.66, 130.59, 130.39, 130.33, 129.08, 128.64, 128.26, 126.67, 126.08, 125.51, 123.33, 122.32, 122.20, 122.16, 118.56, 114.07, 109.46, 55.28, 45.31, 37.09. Recrystallized in EtOH:AcOEt.*1-benzyl-2-((4-methoxybenzyl)thio)-1H-benzo[d]imidazole (***19***)*. **Yield:** 86%. White solid. **^1^H NMR** (400 MHz, Chloroform-*d*) δ 7.76 (d, *J* = 7.9 Hz, 1H), 7.34 (d, *J* = 8.3 Hz, 2H), 7.30–7.22 (m, 4H), 7.17 (d, *J* = 4.3 Hz, 2H), 7.12–7.07 (m, 2H), 6.83 (d, *J* = 8.3 Hz, 2H), 5.24 (s, 2H), 4.61 (s, 2H), 3.79 (s, 3H). **^13^C NMR** (101 MHz, CDCl_3_) δ 159.16, 151.86, 143.71, 136.16, 135.67, 130.34, 128.82, 128.63, 127.88, 126.89, 122.19, 122.06, 118.47, 114.08, 109.31, 55.28, 47.60, 37.14. Recrystallized in H_2_O:EtOH.*1-ethyl-2-((4-methoxybenzyl)thio)-1H-benzo[d]imidazole (***20***)*. **Yield:** 63%. Yellow oil. **^1^H**
**NMR** (400 MHz, DMSO-*d*_6_) δ 7.62–7.56 (m, 1H), 7.51–7.45 (m, 1H), 7.37 (d, *J* = 8.5 Hz, 2H), 7.22–7.13 (m, 1H), 6.86 (d, *J* = 8.5 Hz, 2H), 4.56 (s, 2H), 4.11 (q, *J* = 7.2 Hz, 2H), 3.71 (s, 3H), 1.21 (t, *J* = 7.2 Hz, 3H). **^13^C NMR** (101 MHz, DMSO) δ 158.60, 150.47, 143.06, 135.57, 130.16, 129.10, 121.57, 117.68, 113.87, 109.40, 55.04, 39.52, 38.34, 35.27, 14.42. Purified using preparative plate using Hexane:AcOEt (4:1).

#### 4.1.7. General Procedure for the Synthesis of **4**–**6**, **10**–**12**, and **16**–**17**

Here, 2 mmol of the corresponding aliphatic alcohol and 1.2 equivalents of NaH (as 60% oil disp.) were stirred at room temperature for 30 min under N_2_ atmosphere and dry DMF as solvent. Then, 1 mmol of the corresponding *N*-alkyl-2-chlorobenzimidazol **I**–**III** in DMF was added using a syringe, and the reaction mixture was heated in an oil bath at 130 °C for 72 h. Excess NaH was inactivated with MeOH, and the reaction was poured over ice-cold water and extracted with DCM or AcOEt. The organic phase was dried over Na_2_SO_4_ and the solvent removed in vacuo. Products were purified via column chromatography or preparative plate or recrystallization.

*2-((2-isopropyl-5-methylcyclohexyl)oxy)-1-(naphthalen-1-ylmethyl)-1H-benzo[d]imidazole (***4***)*. **Yield:** 33%. White solid. **^1^H NMR** (400 MHz, Chloroform-*d*) δ 8.33 (d, *J* = 8.1 Hz, 1H), 8.13 (d, *J* = 7.9 Hz, 1H), 8.01 (d, *J* = 8.3 Hz, 1H), 7.86 (d, *J* = 7.9 Hz, 1H), 7.78 (p, *J* = 7.0 Hz, 2H), 7.55 (t, *J* = 7.8 Hz, 1H), 7.39 (t, *J* = 7.5 Hz, 1H), 7.30–7.14 (m, 3H), 5.85 (s, 2H), 5.34 (td, *J* = 10.9, 4.4 Hz, 1H), 2.69 (d, *J* = 11.0 Hz, 1H), 2.17–2.04 (m, 1H), 2.02–1.80 (m, 3H), 1.68 (t, *J* = 11.7 Hz, 1H), 1.47–1.23 (m, 2H), 1.16 (d, *J* = 6.2 Hz, 3H), 1.13–1.05 (m, 1H), 1.02 (d, *J* = 7.0 Hz, 6H). **^13^C NMR** (101 MHz, CDCl_3_) δ 157.32, 140.56, 133.89, 133.76, 131.44, 130.78, 129.02, 128.27, 126.45, 125.96, 125.41, 124.24, 122.58, 121.55, 120.76, 117.66, 108.70, 80.70, 47.66, 43.58, 40.71, 34.34, 31.29, 26.38, 23.54, 22.08, 20.74, 16.66. Purified via preparative plate using Hexane:AcOEt (9:1), then recrystallized in MeOH.*1-benzyl-2-((2-isopropyl-5-methylcyclohexyl)oxy)-1H-benzo[d]imidazole (***5***)*. **Yield:** 46%. White solid. **^1^H NMR** (400 MHz, Chloroform-*d*) δ 7.73 (d, *J* = 7.8 Hz, 1H), 7.50–7.38 (m, 3H), 7.42–7.26 (m, 3H), 7.30–7.19 (m, 2H), 5.37–5.16 (m, 3H), 2.64–2.54 (m, 1H), 2.04 (pd, 1H), 1.94–1.74 (m, 3H), 1.74–1.62 (m, 1H), 1.38–1.16 (m, 2H), 1.14–1.05 (m, 4H), 1.02 (d, *J* = 7.0 Hz, 3H), 0.93 (d, *J* = 7.0 Hz, 3H). **^13^C NMR** (101 MHz, CDCl_3_) δ 157.16, 140.45, 136.48, 133.61, 128.71, 127.66, 127.03, 121.46, 120.67, 117.59, 108.35, 80.52, 47.84, 45.56, 40.69, 34.35, 31.27, 26.35, 23.53, 22.06, 20.81, 16.64. Purified via preparative plate using Hexane:AcOEt (9:1), then recrystallized in MeOH:H_2_O.*1-ethyl-2-((2-isopropyl-5-methylcyclohexyl)oxy)-1H-benzo[d]imidazole (***6***)*. **Yield:** 35%. Yellow solid. **^1^H NMR** (400 MHz, DMSO-*d*_6_) δ 7.41–7.34 (m, 1H), 7.36–7.29 (m, 1H), 7.10–7.02 (m, 2H), 4.94 (td, *J* = 4.3, 10.8 Hz, 1H), 4.00 (q, *J* = 7.2 Hz, 2H), 2.28–2.18 (m, 1H), 2.01 (pd, *J* = 2.7, 6.9 Hz, 1H), 1.75–1.64 (m, 2H), 1.63–1.48 (m, 2H), 1.24 (t, *J* = 7.2 Hz, 3H), 1.21–1.02 (m, 2H), 0.99–0.91 (m, 1H), 0.90 (dd, *J* = 1.9, 6.8 Hz, 6H), 0.77 (d, *J* = 7.0 Hz, 3H). **^13^C NMR** (101 MHz, DMSO) δ 156.80, 140.36, 133.50, 121.27, 120.67, 117.33, 108.98, 80.15, 47.42, 40.69, 36.69, 34.23, 31.32, 26.70, 23.75, 22.34, 20.89, 17.10, 14.79. Purified via preparative plate using Hexane:AcOEt (9:1).*(E)-2-((3,7-dimethylocta-2,6-dien-1-yl)oxy)-1-(naphthalen-1-ylmethyl)-1H-benzo[d]imidazole (***10***)*. **Yield:** 29%. Yellow oil. **^1^H NMR** (400 MHz, Chloroform-*d*) δ 8.11 (d, *J* = 8.2 Hz, 1H), 7.72 (d, *J* = 7.6 Hz, 1H), 7.63 (d, *J* = 8.2 Hz, 1H), 7.48–7.33 (m, 2H), 7.27–7.20 (m, 2H), 7.13 (dd, *J* = 7.1, 1.2 Hz, 1H), 6.76 (pd, *J* = 7.5, 1.5 Hz, 2H), 6.68–6.62 (m, 1H), 6.24 (dd, *J* = 17.6, 10.9 Hz, 1H), 5.47 (d, *J* = 16.1 Hz, 0H), 5.32 (d, *J* = 16.1 Hz, 0H), 5.17–5.08 (m, 2H), 5.00–4.94 (m, 1H), 2.20–2.11 (m, 2H), 2.11–1.95 (m, 1H), 1.82 (s, 3H), 1.81–1.71 (m, 1H), 1.50 (s, 3H), 1.35 (s, 3H). **^13^C NMR** (101 MHz, CDCl_3_) δ 154.50, 144.19, 133.90, 131.95, 131.65, 131.23, 130.03, 129.56, 128.85, 128.41, 126.56, 125.98, 125.27, 125.16, 123.74, 123.26, 120.89, 120.48, 113.15, 112.51, 108.33, 63.64, 42.95, 38.49, 29.77, 25.70, 25.37, 22.95, 17.61.*(E)-1-benzyl-2-((3,7-dimethylocta-2,6-dien-1-yl)oxy)-1H-benzo[d]imidazole (***11***)*. **Yield:** 33%. Yellow oil. **^1^H NMR** (400 MHz, Chloroform-*d*) δ 7.32–7.22 (m, 5H), 7.23–7.14 (m, 1H), 6.95–6.83 (m, 2H), 6.84–6.75 (m, 1H), 6.30 (dd, *J* = 10.8, 17.6 Hz, 1H), 5.25–5.15 (m, 2H), 5.07–4.91 (m, 3H), 2.30–2.14 (m, 2H), 2.08 (dq, *J* = 5.6, 6.2, 11.8 Hz, 1H), 1.88 (s, 3H), 1.86–1.77 (m, 1H), 1.57 (s, 3H), 1.43 (s, 3H). **^13^C NMR** (101 MHz, CDCl_3_) δ 154.59, 144.15, 136.66, 131.88, 129.68, 129.48, 128.78, 128.73, 128.49, 127.55, 127.37, 123.68, 120.87, 120.46, 113.10, 112.53, 107.79, 63.55, 44.41, 38.31, 25.68, 25.30, 22.92, 17.55.*(E)-2-((3,7-dimethylocta-2,6-dien-1-yl)oxy)-1-ethyl-1H-benzo[d]imidazole (***12***)*. **Yield**: 32%. Yellow oil. **^1^H NMR** (400 MHz, Chloroform-*d*) δ 7.31 (d, *J* = 7.8 Hz, 1H, **H_4_**), 7.07–7.01 (m, 1H, **H_1_**), 6.97–6.90 (m, 2H, **H_2_**_–**3**_), 6.31 (dd, *J* = 10.8, 17.6 Hz, 1H, **H_21_**), 5.33–5.15 (m, 2H), 5.04 (t, *J* = 7.2 Hz, 1H), 3.90 (q, *J* = 7.1 Hz, 2H), 2.32–2.14 (m, 2H), 2.13–2.00 (m, 1H), 1.90–1.77 (m, 4H), 1.59 (s, 3H), 1.45 (s, 3H), 1.31 (t, *J* = 7.1 Hz, 3H). **^13^C NMR (101 MHz, CDCl_3_)** δ 144.16, 131.80, 129.47, 123.66, 120.66, 120.06, 112.95, 112.47, 106.94, 63.36, 38.24, 35.45, 25.62, 25.15, 22.84, 17.47, 13.44. Purified via preparative plate using Hexane:AcOEt (9:1).*2-(adamantan-1-yloxy)-1-benzyl-1H-benzo[d]imidazole (***16***)*. **Yield:** 5%. White solid. **^1^H NMR** (200 MHz, Chloroform-*d*) δ 7.56 (dd, *J* = 7.1, 1.5 Hz, 1H), 7.36–7.17 (m, 6H), 7.12 (dd, *J* = 7.7, 4.4 Hz, 1H), 7.08–7.02 (m, 2H), 5.13 (s, 2H), 2.26 (s, 9H), 1.69 (s, 6H). **^13^C NMR** (50 MHz, CDCl_3_) δ 155.16, 140.66, 136.74, 132.70, 128.65, 127.57, 127.16, 121.26, 120.63, 117.86, 108.38, 83.32, 77.64, 77.01, 76.37, 45.73, 41.72, 36.06, 31.09. Purified via preparative plate using Hexane:AcOEt (6:1).*2-(adamantan-1-yloxy)-1-ethyl-1H-benzo[d]imidazole (***17***)*. **Yield:** 21%. White solid. **^1^H NMR** (400 MHz, Acetone-*d_6_*) δ 7.45–7.35 (m, 1H), 7.30–7.22 (m, 1H), 7.10–7.00 (m, 2H), 4.05 (q, *J* = 7.2 Hz, 2H), 2.40–2.32 (m, 6H), 2.27–2.19 (m, 3H), 1.82–1.67 (m, 6H), 1.31 (t, *J* = 7.2 Hz, 3H). **^13^C NMR** (101 MHz, Acetone) δ 155.62, 141.81, 133.38, 121.55, 121.02, 118.18, 108.95, 82.88, 42.28, 37.35, 36.83, 31.97, 14.70. Purified via preparative plate using Hexane:AcOEt (4:1).

#### 4.1.8. General Procedure for the Synthesis of **21**–**26**

Here, 1 mmol of the corresponding aromatic alcohol, 1 mmol of *N*-alkyl-2-benzylsulfonylbenzimidazole **VII**–**IX**, and 1 equivalent of Cs_2_CO_3_ were dissolved in DMF. The reaction mixture was heated in an oil bath at 130 °C for 24 h and was poured over ice-cold water. The aqueous phase was extracted with DCM or AcOEt, the organic phase was dried over Na_2_SO_4_, and the solvent removed in vacuo. Products were purified via preparative plate.

*2-(3-methoxyphenoxy)-1-(naphthalen-1-ylmethyl)-1H-benzo[d]imidazole (***21***)*. **Yield:** 39%. White solid. **^1^H NMR** (400 MHz, DMSO-*d*_6_) δ 8.29 (d, *J* = 8.9 Hz, 1H), 8.00 (d, *J* = 7.7 Hz, 1H), 7.89 (d, *J* = 8.2 Hz, 1H), 7.67–7.56 (m, 2H), 7.50 (d, *J* = 8.3 Hz, 1H), 7.44 (t, *J* = 7.7 Hz, 1H), 7.39–7.32 (m, 2H), 7.20–7.07 (m, 2H), 7.02 (dd, *J* = 7.0, 1.2 Hz, 1H), 6.98–6.92 (m, 2H), 6.86 (ddd, *J* = 8.3, 2.4, 0.9 Hz, 1H), 5.96 (s, 2H), 3.75 (s, 3H). **^13^C NMR** (101 MHz, DMSO) δ 160.76, 155.88, 154.87, 139.86, 134.01, 133.82, 132.38, 130.78, 130.67, 129.19, 128.56, 127.05, 126.65, 125.98, 124.34, 123.54, 122.23, 121.99, 118.48, 112.72, 111.66, 110.36, 106.74, 55.89, 44.07. Purified via column chromatography using gradient elution from DCM to DCM:MeOH 5%, then recrystallized in AcOEt.*1-benzyl-2-(3-methoxyphenoxy)-1H-benzo[d]imidazole (***22***)*. **Yield:** 66%. Yellow oil. ^1^H NMR (400 MHz, Chloroform-*d*) δ 7.66–7.58 (m, 1H), 7.38–7.27 (m, 6H), 7.23–7.16 (m, 3H), 6.95 (ddd, *J* = 8.1, 2.3, 0.9 Hz, 1H), 6.91 (t, *J* = 2.4 Hz, 1H), 6.81 (ddd, *J* = 8.3, 2.4, 0.9 Hz, 1H), 5.34 (s, 2H), 3.80 (s, 3H). **^13^C NMR** (101 MHz, CDCl_3_) δ 160.83, 155.65, 154.63, 139.79, 135.97, 133.34, 130.26, 128.94, 127.99, 127.18, 122.11, 121.75, 118.79, 112.30, 111.52, 109.04, 106.26, 55.48, 46.20. Purified via preparative plate using Hexane:AcOEt (6:1).*1-ethyl-2-(3-methoxyphenoxy)-1H-benzo[d]imidazole (***23***)*. **Yield:** 50%. Yellow oil. **^1^H NMR** (400 MHz, Chloroform-*d*) δ 7.67–7.57 (m, 1H), 7.34 (t, *J* = 8.2 Hz, 1H), 7.31–7.25 (m, 1H), 7.28–7.17 (m, 2H), 7.01–6.92 (m, 2H), 6.82 (dd, *J* = 8.2, 2.3 Hz, 1H), 4.21 (q, *J* = 7.2 Hz, 2H), 3.84 (s, 3H), 1.49 (t, *J* = 7.2 Hz, 3H). **^13^C NMR** (101 MHz, CDCl_3_) δ 160.83, 155.34, 154.80, 139.92, 133.01, 130.24, 121.76, 121.44, 118.79, 112.14, 111.33, 108.44, 106.10, 55.48, 37.43, 14.67. Purified via preparative plate using Hexane:DCM (1:4).*2-(3-methoxy-5-(2-methyloctan-2-yl)phenoxy)-1-(naphthalen-1-ylmethyl)-1H-benzo[d]imidazole (***24***)*. **Yield:** 29%. Yellow oil. **^1^H NMR** (200 MHz, Chloroform-d) δ 8.15 (d, J = 7.3 Hz, 1H), 7.92 (dd, J = 7.7, 1.9 Hz, 1H), 7.81 (d, J = 8.3 Hz, 1H), 7.68–7.48 (m, 3H), 7.37 (t, J = 7.7 Hz, 1H), 7.27–7.00 (m, 4H), 6.87–6.70 (m, 3H), 5.83 (s, 2H), 3.77 (s, 3H), 1.62–1.45 (m, 2H), 1.31–1.13 (m, 16H), 0.87–0.72 (m, 4H). **^13^C NMR** (50 MHz, CDCl_3_) δ 160.27, 155.93, 154.39, 152.99, 140.06, 133.78, 133.68, 131.05, 130.67, 129.07, 128.47, 126.66, 126.08, 125.45, 124.11, 122.42, 122.06, 121.67, 118.84, 110.15, 109.28, 102.39, 77.66, 77.02, 76.39, 65.84, 55.38, 44.45, 44.13, 38.02, 31.75, 29.99, 28.81, 24.61, 22.67, 14.06. Purified via preparative plate using DCM:NEt_3_ (0.1%).*1-benzyl-2-(3-methoxy-5-(2-methyloctan-2-yl)phenoxy)-1H-benzo[d]imidazole (***25***)*. **Yield:** 66%. Yellow oil. **^1^H NMR** (200 MHz, Chloroform-*d*) δ 7.66–7.56 (m, 1H), 7.42–7.24 (m, 5H), 7.25–7.07 (m, 3H), 6.82 (t, *J* = 1.9 Hz, 1H), 6.77 (d, *J* = 1.8 Hz, 2H), 5.34 (s, 2H), 3.79 (s, 3H), 1.65–1.50 (m, 2H), 1.33–1.17 (m, 12H), 1.15–1.02 (m, 2H), 0.96–0.79 (m, 3H). **^13^C NMR** (50 MHz, CDCl_3_) δ 160.43, 155.84, 154.62, 153.15, 140.13, 136.19, 133.51, 129.02, 128.05, 127.29, 122.10, 121.71, 118.93, 110.29, 110.24, 109.10, 102.52, 77.80, 77.16, 76.53, 55.53, 46.30, 44.59, 38.17, 31.90, 30.14, 28.97, 24.76, 22.81, 14.20. Purified using preparative plate using Hexane:AcOEt (6:1).*1-ethyl-2-(3-methoxy-5-(2-methyloctan-2-yl)phenoxy)-1H-benzo[d]imidazole (***26***)*. **Yield:** 49%. **^1^H NMR** (200 MHz, Chloroform-*d*) δ 7.64–7.51 (m, 1H), 7.30–7.11 (m, 3H), 6.85 (t, *J* = 1.9 Hz, 1H), 6.81–6.71 (m, 2H), 4.20 (q, *J* = 7.2 Hz, 2H), 3.79 (s, 3H), 1.65–1.52 (m, 2H), 1.47 (t, *J* = 7.2 Hz, 3H), 1.24 (d, *J* = 11.3 Hz, 12H), 1.10 (dt, *J* = 7.7, 4.0 Hz, 2H), 0.85 (q, *J* = 6.1, 4.9 Hz, 3H). **^13^C NMR** (50 MHz, CDCl_3_) δ 160.45, 155.54, 154.68, 153.14, 140.12, 133.15, 121.82, 121.46, 118.88, 110.17, 108.51, 102.39, 77.79, 77.16, 76.52, 55.53, 44.60, 38.18, 37.52, 31.90, 30.14, 28.99, 24.76, 22.81, 14.81, 14.20.

#### 4.1.9. General Procedure for the Synthesis of Compounds **IVa**–**b**

The synthetic procedure was adapted from previously reported procedures [28]. In brief, 1 equivalent of alcohol, 1.3 equivalent of *p*-TsCl, and 1.5 equivalent of DABCO were dissolved in AcOEt (for menthol derivative) or DCM (for geraniol derivative). The reaction was left to stir overnight forming a white suspension. The crude reaction was filtered in both cases. For menthol derivative, the organic phase was concentrated under vacuum, and an oily residue was obtained, which solidified over time. For geraniol derivative, the organic phase was cooled and a white solid precipitated again and was filtered. The solid corresponded to the product.

### 4.2. Molecular Docking

Molecular docking was performed using the cryoEM structure of the CB2 receptor co-crystallized with agonist WIN-55212-2 (PDB:6PT0). Structure of receptor was prepared for docking using UCSF Chimera software, removing all molecules and chains except the receptor itself. Ligands were submitted to energy minimization using Spartan. Docking was performed using Autodock suite 4.2.6. with a grid box of 48 × 44 × 42 centered at x: 98.857, y: 109.109, and z: 124.164. Files were prepared using AutoDockTools-1.5.7. Docking results were visually analyzed, and relevant binding modes were selected and further analyzed with Protein–Ligand Interaction Profiler (PLIP) [32]. Binding poses were obtained using PyMol software (The PyMOL Molecular Graphics System, Version 2.6.0 Schrödinger, LLC).

### 4.3. cAMP Accumulation Assay

In vitro pharmacology assay for human CB2 agonist effect in CHO cells was performed by Eurofins Cerep (Celle-Lévescault, France). Five compounds were tested at several concentrations for EC50 determination. Cellular agonist effect was calculated as a % of control response to the known reference agonist WIN55212-2 (100 nM). CHO cells stably expressing hCB2 receptor were used to determine agonist effect of compounds, based on the measurement of cAMP level. Assay protocol is adapted from Felder et al. [33].

### 4.4. Cannabinoid Binding Assay

Single-dose binding assays in hCB2R CHO cells were conducted at Eurofins Cerep, France, with the CB2R agonist, [^3^H]WIN55212-2 (0.8 nM, Kd of 1.5 nM). Non-specific binding was defined in the presence of 5 µM WIN55212-2. All membrane preparations for these studies were generated at Cerep Inc. according to standard protocols. Duplicate determinations were performed for each test compound. Compounds were dissolved in DMSO to generate a stock solution. Radioligand binding methods were adapted from Munro et al. [34].

### 4.5. CB1/CB2 Receptor Binding Assay

CB1 and CB2 receptor binding assays were performed exactly as previously described [35]. Briefly, membranes from HEK cells overexpressing the respective human recombinant CB1R (Bmax = 2.5 pmol/mg protein) and human recombinant CB2R (Bmax = 4.7 pmol/mg protein) were incubated with [^3^H]-CP-55,940 (0.14 nM/Kd = 0.18 nM and 0.084 nM/Kd = 0.31 nM, respectively, for CB1R and CB2R) as the high-affinity ligand. Nonspecific binding was defined by 10 μM WIN55 212−2 as the heterologous competitor (Ki values 9.2 and 2.1 nM, respectively, for CB1R and CB2R). Displacement curves were performed by incubating [^3^H]-CP-55 940 (90 min, 30 °C) with increasing concentrations of compounds (10 nM−10 μM). Ki values were calculated by applying the Cheng−Prusoff equation to the IC50 values (obtained by GraphPad, Prism Software 9.5) for the displacement of the bound radioligand by compounds.

### 4.6. Cell Culture

MCF-7 and HEK-293 cell lines were maintained at 37 °C in a humidified atmosphere containing 5% CO_2_. MCF-7 was grown in Roswell Park Memorial Institute (RPMI) 1640 and HEK-293 was grown in Dulbecco’s modified Eagle medium and Ham’s F-12 medium (DMEM/F12). All media was supplemented with 10% fetal bovine serum, penicillin (100 U/mL), streptomycin (100 µM/mL), and non-essential amino acids MEM 1×.

### 4.7. Neutral Red Uptake Assay

Cells were seeded in 100 μL of media at a density of 10^4^ cells/well in 96-well microtiter plates. Solutions of compounds were previously prepared in DMSO, and 1 μL of the corresponding solution was added to each well. The final volume of each well was adjusted to 200 μL. After 72 h of incubation, culture media were removed, and 100 μL of 10 μg/mL neutral red solution prepared in culture media was added to each well and incubated for 2 h at 37 °C in a humidified atmosphere containing 5% CO_2_. Then, media were aspirated, the plate was washed three times with PBS 1X, and 100 μL of neutral red distain solution (50:49:1, ethanol/water/glacial acetic acid) was added. The plate was placed for 15 min in a shaker, and fluorescence was measured using Cytation 5 apparatus (Biotek, Winooski, VT, USA) at 530/645 nm excitation/emission wavelengths.

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
