# Peer review of "Motifs in Natural Products as Useful Scaffolds to Obtain Novel Benzo[d]imidazole-Based Cannabinoid Type 2 (CB2) Receptor Agonists"

_ijms, 2023, doi:10.3390/ijms241310918_

Round 1

Reviewer 1 Report

The idea of this work was to combine benzimidazole scaffold with natural products parts to obtain compounds with expected activity toward cannabinoid CB2 receptor. Obtained were 26 new compounds as the result of introduction of substituents in positions 1- (naphthyl, benzyl, ethyl) and 2- (natural structures motifs) of benzimidazole scaffold. Obtained compounds were tested for activity toward cannabinoid CB-2 receptor in radioligand binding studies. Five most active compounds were tested for selectivity toward cannabinoid  CB-1 receptor and in cAMP accumulation assay.  Molecular modeling studies were performed to analyze binding modes of three most active compounds in orthosteric pocket of CB2 receptor model. Toxicity of compound named 6 was examined for HEK-293 and MCF-7 cell lines. This work is reasonably planned but at the present form is not suitable for publication in IJMS because:

 Experiments are not described in proper, detailed way, obtained results of biological tests are without statistical data, presented with only one value. The source of used cell lines was not provided. The description of synthetic experiments are without details (characteristic of obtained compounds), only general descriptions were given. Unfortunately I was unable under given address- www.mdpi.com/xxx/s1 to achieve supporting information

Author Response

Response 1: We thank reviewer 1 for his comments. The supporting information file has been corrected and characterization of the synthesized compounds has been included in de section 4. Materials and Methods; 4.1 Chemistry. Statistical data and cell lines have been included in the functional and receptor selectivity assays (Figure 4 and Table 4 in manuscript).

Reviewer 2 Report

In the manuscript titled "Motifs in natural products as useful scaffolds to obtain novel 2 benzo[d]imidazole-based cannabinoid type 2 (CB2) receptor ag-3 onists", author describe the synthesis a series of CB2 ligands. The presentation of this work is really very nice. The author has systematically studied the biological activities by the radioligand displacement assay, cAMP accumulation assay and molecular docking. Some of synthesized compounds show high potency with low toxicity. I recommended to accept this manuscript in the present form. 

Unfortunately, I do not find the supporting information. Please make sure it is uploaded and contain all the required analytical data.

Author Response

We thank reviewer 2 for his comments. The supporting information file has been corrected and characterization of the synthesized compounds has been included in de section 4. Materials and Methods; 4.1 Chemistry.

Reviewer 3 Report

Analia et al. are presenting an interesting study of medicinal chemistry research to report CB2 agonist. The manuscript is comprehensively composed with chemistry, biological competitive binding and functional validations, and computational assessment. Overall, the work is well-presented. The following are specific comments and questions.

1. The radioligand displacement assay gave a single percentage inhibition data point for each compound in Table 2. The assay can possibly be repeated to report the mean and stdev.

2. For the functional validation, the cAMP assay was used. Would it be beneficial to have the orthogonal validation with beta-arrestin and GTPgS assays?

3. In CB1/CB2 selectivity determination, again, should the competitive binding Ki values be measured repeatedly to report the mean and stdev?

4. In CB1/CB2 selectivity determination, Ki CB1 for compounds 6 and 16 are >10 uM. Then, how does the Ki(CB1)/Ki(CB2) be calculated?

5. In Result section 2.6, molecular docking, authors simply said they performed the docking. But what is the result? What are key receptor-ligand interactions? How these interactions differ from WIN compound? How about using another antagonist-bound CB2 model to dock for comparison? A more detailed results are expected.

6. In discussion, authors talked about some observed interactions in docking. MD simulations are recommended to be continued to evaluate if observed interactions can be stably maintained. 

7. What are next research steps for these identified novel CB2 agonists? What are further modification directions? These questions can be briefly touched in the discussion part. Besides, some in silico ADME properties prediction can be a beneficial addition. 

Author Response

We thank reviewer 3 for his comments.

Point 1: Radioligand displacement assay were performed in duplicate by Eurofins Cerep services as indicated in section 4. Materials and Methods; 4.4 Cannabinoid Binding assay

Point 3: In CB1/CB2 selectivity determination 3 independent experiments run in duplicate were performed. Further details and a reference have been included in the methods section (line 447) and statistical data has been added in Figure 4 and Table 4.

Point 4: Calculation of the selectivity index was performed using 10 uM as the lowest possible value of Ki CB1. Values have been corrected in Table 4.

Point 5: More detailed discussion of the docking results is provided in the 3. Discussion section, lines 277-299.

Points 2, 6 and 7: Further characterization of the specific profiles of the identified compounds is certainly necessary and is a subject of ongoing work in our laboratory. Specifically, pharmacokinetic and pharmacodynamic properties through the interaction with human serum albumin (HAS) are current next steps in our research. Orthogonal validation with additional functional assays will also be useful as well as molecular dynamics studies to analyze the stability and binding interactions.

Reviewer 4 Report

The subject matter is somewhat relevant to the readers of the journal, and the author has emphasized the ways in which the recent research may have an effect on the field. In overall, I thought the manuscript was extremely well written. Each of the parts was understandable and to the point, and there was a decent flow from one paragraph to the next. The title is unambiguous and captures the essence of the research. The work that was completed is summarized in a succinct manner in the abstract. The introduction gives the reader enough background knowledge to make informed decisions, and it is well-structured and backed with data and citations. The Conclusion elucidates in an appropriate manner the advantages of the research.

However, some major comments need to be addressed.

1. Why did you choose these two HEK-293 and MCF-7 cell lines? For the pharmacological characterization of the newly synthesized compounds, I recommend using normal cells and macrophages.

2. What do you think about synthesizing compounds' interaction with immune cells? Because it is an important concern for the safety of the newly synthesized compounds.

3. What are the physical characteristics of these compounds? Are they well soluble?

4. How do you consider them drug-like compounds except for molecular docking explanation?

5. What about there MW?

Author Response

We thank reviewer 4 for his comments.

Point 1: HEK-293 cell lines are non-cancerous and were selected to represent normal healthy cells. MCF-7 cell lines were selected as a cancerous model to test possible antitumoral activity of the compounds considering the reported relation between CB2 agonists and breast cancer cells MCF-7.1,2

Point 2: Interaction with immune cells is certainly important as CB2 receptors play a central role regulating immune function. Here, toxicity studies have been carried out in models of cancerous and non-cancerous cells as an initial assessment but further evaluation in cell models more specific to their CB2 agonist action such as macrophages and lymphocytes will be important to evaluate.

Point 4: The drug-like nature of the synthesized compounds is based on the combination of two scaffolds, a benzo[d]imidazole heterocycle and a natural product motif, both of which are generally associated to medicinal properties. The benzo[d]imidazole scaffold has been considered a privileged scaffold in drug discovery and is present in various molecules with pharmacologic properties3–7. On the other hand, natural products are commonly associated to antioxidant, antibacterial and anti-inflammatory properties8–10 with many of them designated as Generally Recognized as Safe (GRAS) substances by the FDA.

Additionally, the synthesized compounds were analyzed in the web tool Swissadme (http://www.swissadme.ch/) for preliminary in silico evaluation of pharmacokinetics, druglike nature and medicinal chemistry friendlines which predicted good drug-like properties satisfying the four druglikness filters, Lipinsky, Ghose, Veber and Egan.

Points 3 and 5: The supporting information file has been corrected and characterization of the synthesized compounds has been included in de section 4. Materials and Methods; 4.1 Chemistry.

  1. Sophocleous A, Marino S, Logan JG, Mollat P, Ralston SH, Idris AI. Bone cell-autonomous contribution of type 2 cannabinoid receptor to breast cancer-induced osteolysis. J Biol Chem. 2015;290(36):22049-22060. doi:10.1074/jbc.M115.649608
  2. Pagano C, Navarra G, Coppola L, Bifulco M, Laezza C. Molecular mechanism of cannabinoids in cancer progression. Int J Mol Sci. 2021;22(7). doi:10.3390/ijms22073680
  3. Tahlan S, Kumar S, Narasimhan B. Pharmacological significance of heterocyclic 1H-benzimidazole scaffolds: a review. BMC Chem. 2019;13(1). doi:10.1186/s13065-019-0625-4
  4. Keri RS, Hiremathad A, Budagumpi S, Nagaraja BM. Comprehensive Review in Current Developments of Benzimidazole-Based Medicinal Chemistry. Chem Biol Drug Des. 2015;86(1):19-65. doi:10.1111/cbdd.12462
  5. Maruthamuthu, Rajam S, P. CRS, G. BDA, Ranjith R. The chemistry and biological significance of imidazole, benzimidazole, benzoxazole, tetrazole and quinazolinone nucleus. J Chem Pharm Res. 2016;8(5). http://www.jocpr.com/abstract/the-chemistry-and-biological-significance-of-imidazole-benzimidazole-benzoxazole-tetrazole-and-quinazolinone-nucleus-6588.html. Accessed August 16, 2018.
  6. Bansal Y, Silakari O. The therapeutic journey of benzimidazoles: A review. Bioorg Med Chem. 2012;20(21):6208-6236. doi:10.1016/j.bmc.2012.09.013
  7. Ajani OO, Aderohunmu D V., Ikpo CO, Adedapo AE, Olanrewaju IO. Functionalized Benzimidazole Scaffolds: Privileged Heterocycle for Drug Design in Therapeutic Medicine. Arch Pharm (Weinheim). 2016;349(7):475-506. doi:10.1002/ardp.201500464
  8. José Serrano Vega R, Campos Xolalpa N, Josabad Alonso Castro A, Pérez González C, Pérez Ramos J, Pérez Gutiérrez S. Terpenes from Natural Products with Potential Anti-Inflammatory Activity. In: Terpenes and Terpenoids. IntechOpen; 2018. doi:10.5772/intechopen.73215
  9. Hahn D, Shin SH, Bae JS. Natural antioxidant and anti-inflammatory compounds in foodstuff or medicinal herbs inducing heme oxygenase-1 expression. Antioxidants. 2020;9(12):1-40. doi:10.3390/antiox9121191
  10. Del Prado-Audelo ML, Cortés H, Caballero-Florán IH, et al. Therapeutic Applications of Terpenes on Inflammatory Diseases. Front Pharmacol. 2021;12. doi:10.3389/fphar.2021.704197

Reviewer 5 Report

Manuscript ID: ijms-2334203

          The article titled “Motifs in natural products as useful scaffolds to obtain novel benzo[d]imidazole-based cannabinoid type 2 (CB2) receptor agonists” by Carlos David Pessoa-Mahana et al have demonstrated the synthesis of 26 new compounds targeting the CB2 receptor based on combination of different natural product motifs with the benzo[d]imidazole scaffold. From this subset of compounds 15 presented CB2 binding and 3 showed potent agonist activity. The paper deals with development of benzo[d]imidazole-based CB2-selective ligands without interaction with CB1 activation and preventing the associated psychoactive effects.

In the abstract, include the EC50 value of pharmacologic activity and IC50 of toxicity in parenthesis for compound 6 to highlight the significant difference observed.

In figure 1, the general structure of benzo[d]imidazole scaffold or one of the active compound derived from this study could be placed in similar fashion to enforce the design inspired by the structure of tetrahydrocannabinol (THC). The aliphatic/aromatic cores could be also highlighted in this potent molecule.

The impressive synthesis of all the molecules has been accomplished using various strategies clearly defined in the chemistry section. But I didn’t find the NMR and MS data in the manuscript. The supplementary file/data is missing. All the yields of the final or intermediate products should be included in the table 1 using parenthesis or provided in the supplementary file.

The radio-ligand binding assay was used to assess synthesized molecules binding affinity to the CB2 receptors. More than 50% of the molecules provided >50% displacement, while few had agonist activity with >80% inhibition.

In figure 2, use of designated numbers for these molecules in Table 1 or 2 instead of the shown substituents on the X-axis can simplify the clutter.

In the discussion, authors have mention ref no-16 for previous QSAR study using the similar benzo[d]imidazole molecules, while at line 90-91 of manuscript, for same benzo[d]imidazole ref-13 is indicated which describes QSAR for Aminoalkylindoles. This should be corrected accordingly. Correct references is the full responsibility of the authors which should not mislead the readers of IJMS.  

Although the authors have exhaustively demonstrated the usefulness of these new molecules for CB2/CB1 activity/selectivity along with the toxicity study for the most potent molecule 6, most of the testing was done in-vitro.

For the future studies, they mention different assays, which should be specified. This would point the right direction for future work especially the in-vivo & bioavailability data is crucial for further development. 

Author Response

We thank reviewer 5 for his comments.

Point 1: The EC50 value of pharmacologic activity and IC50 of toxicity of compound 6 were included in the abstract (line 31).

Point 2: The general structure of the most active compound was included in figure 1.

Point 3: The supporting information file has been corrected and characterization of the synthesized compounds has been included in de section 4. Materials and Methods; 4.1 Chemistry.

Point 4: This data was clarified in lines 152-154 as suggested.

Point 5: The X-axis in figure 2 has been modified as suggested.

Point 6: Reference numbers have been corrected (line 91).

Round 2

Reviewer 1 Report

Revised manuscript was corrected according to the reviewer's suggestions.

I recommend to publish this mnuscript in present form

Reviewer 3 Report

Authors have appropriately addressed reviewer's comments and questions. The reviewer believes this is a well-prepared paper that can draw broad interests in research communities on cannabinoid receptors. 

Reviewer 4 Report

Authors have answered all the queries. I recommend accepting the manuscript.

Reviewer 5 Report

Accept